# Analysis of Cadmium, Mercury, and Lead Concentrations in Erythrocytes of Renal Transplant Recipients from Northwestern Poland

**DOI:** 10.3390/biology10010062

**Published:** 2021-01-16

**Authors:** Aleksandra Wilk, Maciej Romanowski, Barbara Wiszniewska

**Affiliations:** 1Department of Histology and Embryology, Pomeranian Medical University in Szczecin, al. Powstańców Wielkopolskich 72, 70-111 Szczecin, Poland; barbara.wiszniewska@pum.edu.pl; 2Department and Clinic of General Surgery and Transplantation, Medical University of Warsaw, ul. Nowogrodzka 59, 02-006 Warsaw, Poland; macroma39@gmail.com

**Keywords:** cadmium, lead, mercury, immunosuppressive treatment, renal graft, transplant recipient

## Abstract

**Simple Summary:**

The aim of this study is to determine the blood erythrocyte concentrations of toxic metals (Cd, Pb, and Hg) in renal transplant recipients. Additionally, we analyzed the effect of selected biological and environmental factors, including the intake of various immunosuppressive drug regimens and smoking, on these xenobiotic concentrations. In summary, our data suggest that, smoking is associated with Pb and Cd concentrations, and gender, age change depending on Pb concentration in erythrocytes of renal transplant recipients. Additionally, this is the first research that suggests that immunosuppressive regimen, depending on type of immunosuppressive drugs combination affects Pb concentration in erythrocytes of the mentioned group of patients. It seems to be crucial information for patients who use immunosuppressive drugs.

**Abstract:**

Cadmium (Cd), mercury (Hg), and lead (Pb) exhibit highly nephrotoxic properties, and their high concentrations can lead to renal failure. Much research has been conducted on the concentrations of heavy metals, microelements, and macroelements in the blood, but little is known about the concentration of Cd, Pb, and Hg in erythrocytes of renal recipients. The aim of this study is to determine the blood erythrocyte concentrations of toxic metals (Cd, Pb, and Hg) in renal transplant recipients (RTRs). Additionally, we analyzed the effect of selected biological and environmental factors, including the intake of various immunosuppressive drug regimens and smoking, on these xenobiotic concentrations. The material consisted of erythrocyte samples from 115 patients of the Department of Nephrology, Transplantology, and Internal Medicine at Independent Public Clinical Hospital No. 2, Pomeranian Medical University, Szczecin, in northwestern Poland. Cd, Hg, and Pb levels in the erythrocytes were quantified by inductively coupled mass spectroscopy (ICP-MS). Equal concentrations of Cd were found in erythrocytes of both female and male transplant recipients. The highest level of Hg was seen in women, and women overall had statistically higher concentrations of Pb than men. Comparison of metal concentrations between those over 50 years and those under it showed that Pb concentration was also significantly higher in renal transplant recipients over 50. Pb concentration was almost twice as high in RTRs who used tacrolimus with mycophenolate mofetil than in RTRs who used cyclosporine A with mycophenolate mofetil. The highest level of Cd was seen in smokers, who had 3.25 µg/L. This value was significantly higher than in ex-smokers (*p* = 0.001) and with RTRs who had never smoked. There were significantly higher levels of Pb in the erythrocytes of RTRs who were ex-smokers than in those who had never smoked. A statistically significant correlation was found between Cd and Pb concentrations. Additionally, we have noticed significant positive correlation between Pb and age (R = 0.37), gender (R = 0.24) and significant negative correlation of Pb with GFR (R = −0.33). We have also found significant positive correlation between Hg and age (R = 0.21). In summary, our data suggest that, smoking is associated with Pb and Cd concentrations, and gender, age change depending on Pb concentration in erythrocytes of RTRs. Additionally, this is the first research that suggests that immunosuppressive regimen, depending on type of immunosuppressive drugs combination affects Pb concentration in erythrocytes of RTRs. It seems to be crucial information for patients who use immunosuppressive drugs.

## 1. Introduction

Excessive exposure to heavy metals and their compounds can lead to various pathological alteration within tissues and organs. These elements are distributed in the natural environment and people are exposed to them through ingestion, inhalation, and dermal absorption. Patients with lowered immunity, including renal transplant recipients, are extremely vulnerable to these substances, especially as their immune systems are disturbed by immunosuppressive drugs.

Cadmium (Cd), mercury (Hg), and lead (Pb) are known as the “toxic trio” due to their toxic properties. These xenobiotics are highly nephrotoxic and high concentrations of them can lead to renal failure. Cd is a highly reactive metal whose long half-life (15–20 years) may cause chronic intoxication manifesting as increased urinary albumin and serum creatinine, glycosuria, and phosphaturia [1,2,3,4]. Chronic lead exposure and accumulation in the body can also negatively affect health. High concentrations of Pb can lead to increased blood pressure, which is associated with cardiovascular diseases and appears along with renal disorders [5,6,7]. It is known that inorganic mercury causes immune reactions that lead to glomerular damage, nephrotic syndrome, a reduction in renal flow and glomerular filtration rate, and hypoperfusion leading to acute renal failure [8,9,10]. Methylmercury (MeHg), synthesized by aquatic and soil microorganisms, presents a serious risk to human health; this form of mercury is found in a variety of fish and seafood and is almost 100% absorbed from the human digestive tract; in contrast, only a few percent of metallic mercury and inorganic mercury compounds are absorbed. Furthermore, all three metals exhibit carcinogenic effects [11,12,13]. According to a report by the International Agency for Research on Cancer, Cd is in the group of greatest confirmed risk (1), Pb and Pb inorganic compounds, along with methylmercury, are in Group 2B, while metallic Hg and inorganic Hg compounds are in Group 3.

Concentration of blood chemical elements, including toxic metals, depends on various endo- and exogenous factors, i.e., age, gender, diet, smoking, place of residence. Mean blood concentrations of cadmium and lead increase with age [14,15,16]. Sex differences are also noticeable [15]. Additionally, data on toxic metals show that the industrialization increases the levels of toxic metals in blood [14,16,17]. The blood Cd level across the European cities ranged from 0.25 μg/L in northern Sweden to 0.65 μg/L in Wrocław, Poland [18], blood Pb concentration across European cities varied from 13.4 μg/L in northern Sweden to 26.9 μg/L in Ljubljana, Slovenia, and Hg level in blood across the European cities ranged from 0.40 μg/L in Koprivnica, Croatia to 1.38 μg/L in Northern Sweden [18].

Having entered the bloodstream, most of metals bind to morphotic elements, especially erythrocytes (RBCs). It is known that red blood cells bind 98–99% of lead and 95% of methylmercury [16]. Toxic metals, such as Cd, Hg and Pb accumulate inside the red blood cells. These metals exhibit high affinity to cellular membrane, which is composed of phospholipids. Therefore, we analyze their concentrations in erythrocytes, since they are crucial and extremely reliable marker for detecting of metals regarding the blood. Of note, absorption of heavy metals by erythrocytes may be associated with anion exchangers [19].

Furthermore, the oxidative stress induced by toxic metals may be one of the mechanisms responsible for several liver and kidney diseases. The organelle that plays an important role in the formation of ROS (reactive oxygen species) is mitochondria. High exposition to heavy metals may lead to dysfunction of mitochondria and finally to overproduction of ROS [20]. It is also known that cadmium induces various epigenetic changes in mammalian cells, causing pathogenic risks and the development of various types of cancers. The epigenetics present themselves as chemical modifications of DNA and histones that alter the chromatin without changing the sequence of the DNA nucleotide [20].

Many research centers have conducted studies on the concentration of heavy metals, microelements, and macroelements in blood; however, little is known about the concentrations of Cd, Pb, and Hg found in erythrocytes of renal recipients. Monitoring the concentration of chemical elements, including toxic elements, would seem to be reasonable, as too high blood levels of them may decrease the life span of the transplanted organ. The acceptable blood level of lead in adult humans is less than 10 μg/dL, while the acceptable blood level of cadmium is 0.03–0.12 μg/dL [21]. The mercury concentration in whole blood is usually lower than 10 μg/L, but the value of 20 μg/L or below is considered normal. The blood mercury concentration can rise to 35 μg/L after long-term exposure to mercury vapor [22]. Renal transplant recipients need to use immunosuppressive drugs (IDs) in order to avoid graft rejection, but these IDs affect magnesium (Mg), sodium (Na), [23], and iron (Fe) status in blood [24], as well as that of selenium (Se), copper (Cu), zinc (Zn), and Fe [25]. Our previous data have indicated that mycophenolate mofetil alters the concentration of arsenic and selenium concentration in blood of renal transplant recipients [26]. Arsenic, like Cd, Pb, and Hg, is a toxic element. There is no data on the effects of immunosuppressive drugs on the blood erythrocyte concentrations of Cd, Pb, and Hg. We hypothesize that IDs affect toxic metal concentrations in the erythrocytes of renal transplant recipients, since our previous data indicated altered levels of microelements and arsenic in the same group of patients.

The aim of this study is to determine the blood erythrocyte concentrations of toxic metals (Cd, Pb, and Hg) in renal transplant recipients.

We additionally analyzed the effect of selected biological and environmental factors, including intake of various immunosuppressive drug regimens, smoking, biochemical parameters, on the xenobiotic concentrations. Regarding age, we have chosen 50-year-old as the cut-off to perform the statistical analysis, since it has been shown that the Cd concentration in the kidney, of people over 50 years of age increases and then decreases again after 60–70. Metabolic differences, including menopause and andropause, after 50 years of age appears and this seems to be the optimal cut-off point.

## 2. Materials and Methods

### 2.1. Study Population

The study was approved by the Bioethics Committee of the Pomeranian Medical University (decision KB-0012/74/17). The research was carried out from 2017 to 2019. The material consisted of erythrocyte samples from 115 kidney recipients of the Department of Nephrology, Transplantology, and Internal Medicine at Independent Public Clinical Hospital No. 2, Pomeranian Medical University, in Szczecin in northwestern Poland. The experiment is the continuation of our previous experiment [25,26]. Patients were characterized by stable function of renal graft for over six months. The blood samples were obtained from women aged from 24 to 71 years and from men aged from 28 to 72 years, since these were patients of the Department of Nephrology, Transplantology and Internal Medicine Independent Public Clinical Hospital No. 2, Pomeranian Medical University, who had undergone renal transplantation. The patients were divided in aspect of age: over and under 50 years old, gender, smoking aspect: S—smokers, EXS—ex-smokers, NS—never-smoking patients. Some renal transplant recipients receive combination of two drugs, and some of them are treated with protocol based on three drugs; therefore, we analyzed the results in two aspects, depending on number and type of drugs included in regimen.

### 2.2. Elements Assay

Cd, Hg, and Pb levels in erythrocytes were quantified by inductively coupled mass spectroscopy (ICP-MS NexION 350D, PerkinElmer, Waltham, MA, USA) using methane to reduce polyatomic interferences, following Lubinski et al. [27]. Calibration standards were prepared by diluting 10 mg/L Multi-Element Calibration Standard 3 (PerkinElmer Pure Plus, PerkinElmer Life and Analytical Sciences) with a blank reagent consisting of a 0.65% solution of nitric acid (Merck, Germany) and 0.002% Triton X-100 (PerkinElmer). Calibration curves were created using four different concentrations, namely 0.1 μg/L, 0.5 μg/L, 1 μg/L, and 2 μg/L. Germanium (PerkinElmer Pure, PerkinElmer Life and Analytical Sciences, Waltham, MA, USA) was used as an internal standard and ClinChek Plasma Control Level I (Recipe, Germany) was used as a reference material, measured after each of the six samples: where the difference of the reference material measurements exceeded 5%, the entire series was repeated. Each sample was measured in duplicate in different analytical runs. Prior to analysis, all samples were centrifuged (6000 rpm, 15 min) and the supernatant was diluted one hundred times with the reagent blank. Technical details, plasma operating settings, and mass-spectrometer acquisition parameters are available on request.

### 2.3. Statistical Analysis

The statistical analysis employed StatSoft Statistica 13.3 software and Microsoft Excel 2015. Arithmetic mean (AM), standard deviation (SD), median (Med), and percent coefficient of variation (CV) were established for the concentrations of Cd, Hg, and Pb. To evaluate the compliance of the results with the expected normal distribution, Kolmogorov–Smirnov (KS) tests with the Lilliefors correction were used (*p* < 0.05). In addition, the mean concentrations of the toxic metals in erythrocytes were compared and contrasted between different patient groups. As the data distribution was not consistent with the expected normal distribution, the nonparametric Mann–Whitney U-test was used (MWU; *p* < 0.05). Spearman’s rank correlation coefficients (rs) were also determined. Descriptive analyses and comparison between the levels of tested metals between separate groups was performed. The relationships between metals and between metals and biological, biochemical parameters, and drug concentration were evaluated with Spearman R correlation factors. Significance level for difference test and correlation factor was established at 0.05. 

## 3. Results

### 3.1. Cd, Hg, and Pb Concentrations in All Patients

The concentrations of xenobiotic metals in the erythrocytes of all patients can be arranged in the following descending series: Pb > Cd > Hg. Equal concentrations of Cd were found in the erythrocytes of both female and male transplant recipients (Table 1). The highest level of Hg was noted in women, and women had overall a statistically higher concentration of Pb than men (*p* = 0.008). Comparison of metal concentrations between those over and those under 50 years of age showed that Pb concentration was also significantly higher in renal transplant recipients over 50 (*p* = 0.00003) (Table 2).

### 3.2. Cd, Hg and Pb Concentrations in All Patients in Aspect of Immunosuppressive Regimen

Additionally, we analyzed Cd, Pb, and Hg in erythrocytes of renal transplant recipients grouped by immunosuppressive regimen (Table 2). Drug concentration information was available for cyclosporine A and tacrolimus, and were 86.68 ± 24.06 and 6.6 ± 2.76, respectively. Statistical analysis in the form of the MW U-test showed that the concentration of Pb was almost twice as high in RTRs who used tacrolimus with MMF than in RTRs who used cyclosporine A with MMF (*p* = 0.002) (Table 2). Statistical comparison of metal concentrations the between regimens based on three drugs showed no significant differences, but the level of Pb in the erythrocytes of RTRs whose regimen was based on Tac, MMF, and G was twice as high as in those RTRs whose regimen was based on CsA, MMF, and G (*p* = 0.055). The maximum Pb concentrations of 163.56 µg/L were found in both the three-drug regimens and the two-drug regimens based on tacrolimus (Table 3).

### 3.3. Cd and Pb Concentrations in all Patients in Aspect of Smoking

Comparing the median concentrations of Cd and Pb in erythrocytes between smokers, non-smokers, and ex-smokers, we found that the highest level of Cd was seen for smokers, and was 3.25 µg/L. This value was significantly higher than for ex-smokers (*p* = 0.001) and for RTRs who had never smoked (*p* = 0.000008) (Table 3). Furthermore, we found significantly higher Pb levels in the erythrocytes of RTRs who were ex-smokers than in those who had never smoked (*p* = 0.001) (Table 4).

### 3.4. Correlations between Cd, Hg, and Pb Concentration and Age, Gender, and Biochemical Parameters in Erythrocytes of Renal Transplant Recipients

A statistically significant positive correlation was found between Cd and Pb concentrations (R = 0.19). Additionally, we have noticed significant positive correlation between Pb and age (R = 0.37), gender (R = 0.24) and significant negative correlation of Pb with GFR (R = −0.33). We have also found significant positive correlation between Hg and age (R = 0.21). We have not found any significant correlation between dose of drug and the concentration of tested metals.

## 4. Discussion

We observed that Pb showed the greatest blood erythrocyte concentration of the three toxic metals in the renal transplant recipients; Cd and Hg had generally similar levels to each other, though this depended on the analytical approach. Comparing our results with acceptable norms and other results from Poland, Cd level was increased. Our results regarded patients after renal transplantation. These patients come from the same region of Poland. Perhaps IDs affect Cd concentration in erythrocytes. There is little information in the literature regarding the concentrations of xenobiotic elements in the erythrocytes of transplant recipients, however we sum up data regarding concentrations of toxic metals in blood in other studies (Table 5). Unfortunately, we have not found data on Cd, Pb, and Hg in renal transplant recipients, therefore, we show the levels of the tested elements in blood in different patients, from different countries.

Olszowski et al. [29] indicated that women from northwestern Poland (like in our research) had an average of 0.61 µgCd/L—much less than the renal recipients in our study. Data on blood of patients from other parts of Poland who did not use immunosuppressive drugs also indicated lower blood Cd level than we found. The renal transplant recipients also had lower Hg concentrations than patients from southern Poland and other countries (Table 5); this is likely to be related to the fact that southern Poland is a highly industrialized and anthropogenically transformed region. It is noteworthy that patients from northwestern Poland had higher Pb concentrations than patients from Southern Poland, as reported by Janicka et al. [16], and patients from Spain [8], Mexico [30], and China [31]. It seems that immunosuppressive drugs affect blood xenobiotic levels, as was documented in our previous study on renal tissue [32], where we examined the levels of xenobiotic metals in the erythrocytes of renal transplant recipients, grouped by immunosuppressive regimen. The mechanism of action of immunosuppressive drugs is unclear, but it is known to be associated with oxidative stress [33,34,35]. Interestingly, in blood erythrocytes of renal transplant recipients from Szczecin, we observed significantly higher Pb level in patients treated with mycophenolate mofetil with tacrolimus than in patients who used mycophenolate mofetil together with cyclosporine A (*p* = 0.002). This tendency was also noticed for three-drug regimens: in patients treated with mycophenolate mofetil together with tacrolimus and glucocorticosteroids, blood erythrocytes showed almost twice as high concentrations of lead as in patients who used mycophenolate mofetil with cyclosporine A and with glucocorticosteroids; however, the difference was not statistically confirmed (Table 3). Tacrolimus is said to be more toxic than cyclosporine A, and is used in the treatment of acute rejection in renal transplant recipients. Tacrolimus exhibit hepatotoxic and nephrotoxic properties, and it seems that this calcineurine inhibitor increases the level of Pb in blood.

Sex and age both seem to be crucial for the concentration of toxic metals. Blood levels of lead and cadmium depend on age and sex [15,36]. Our data indicate that males had significantly higher Pb concentrations than females after renal transplantation, at 44.23 µg/L and 35.18 µg/L, respectively. In the study of Garcia-Esquinas et al. (2013), males also showed higher Pb levels than females [8]. Data on Pb concentrations in erythrocytes show that, among RTRs, males had higher Pb levels than females; this is probably associated with greater exposure to this metal; further, men have higher hematocrit levels, as lead binds to erythrocytes [32]. Our data show that patients over 50 had significantly higher Pb levels in erythrocytes than younger patients. Janicka et al. (2015) also reported that Pb concentrations in the blood of older men were twice that of younger men [16]. We noted no significant differences between blood Cd and Hg levels between age groups; however, results from southern Poland indicate that blood Hg levels are higher in the younger group than in the older group, and that Cd concentration did not differ between age groups [16].

Research on patients all over the world indicate various harmful effect of cigarette smoking on a range of tissues and organs [32,37,38,39]. The metal content of tobacco ranges from less than 1 μg/g (for Cd and Pb) to several hundred μg/g (as in the case of aluminum (Al), manganese (Mn), and barium (Ba)). According to Pinto et al. (2017), the highest transfer rate from tobacco to cigarette smoke was found for Tl (85–92%) and Cd (81–90%), followed by Pb (46–60%) and As (33–44%) [40]. In our study, the smoking renal transplant recipients from northwestern Poland had significantly higher Cd in blood erythrocytes than the ex-smokers and non-smokers. Additionally, significantly higher blood Pb was seen in ex-smoker RTRs than in those who never smoked.

Our data confirm the positive correlation between Cd and Pb in the erythrocytes of renal transplant recipients. Both these xenobiotics are strongly nephrotoxic. Renal transplant recipients use immunosuppressive drugs which, as mentioned above, also exhibit nephrotoxic properties. Furthermore, significant positive correlation between Pb and age and gender correlate with other researchers. Finally, negative correlation between Pb and GFR confirms nephrotoxic properties of this metal. Regarding Hg, it accumulates with age, what was noticed in our study.

It is of note, that results regarding toxic metals concentrations in erythrocytes of RTRs were not as high to lead to kidney injury. However, Cd concentration was higher than acceptable norm. No of the patients suffered from intoxication, since their creatinine level was 1.61 ± 0.88, GFR was 53.63 ± 22.41, ALT was 22.55 ± 13.51.

The literature does not seem to have any data on analysis of concentrations of toxic metals in erythrocytes of RTRs; such studies should therefore be expanded. The data presented here confirms that intrinsic factors, including sex and age, affect xenobiotic blood concentrations. Cigarette smoking affects cadmium and lead levels in blood. The cadmium, lead, and mercury profiles of blood erythrocytes in renal transplant recipients from northwestern Poland differs from those of other groups of patients, which may be the result of immunosuppressive drug intake.

The limitation of the study was the small number of patients, therefore, the research should be expanded in different group of patients from the same area. It is also difficult to compare our results with others, since to the best of our knowledge, no data regarding toxic elements in erythrocytes of renal transplant recipients form Poland are available.

Research concerning toxic metals concentration in erythrocytes of organ recipients undoubtedly expand knowledge in nephrotoxicology, and transplantology. Immunosuppressive drugs, although, are necessary for RTRs, they show side effects that patients usually do not know about. These drugs disarm the morphology and function of the organs, and perhaps affect the profile of Cd, Pb, and Hg in erythrocytes. Hence, it also seems reasonable to inform patients about it, especially, in chronic use of immunosuppressive drugs. The application of the optimal treatment scheme and the choice of the appropriate diet are extremely important for RTRs, since it can prolong the proper function of the transplanted organ.

## 5. Conclusions

Our data suggest that, smoking is associated with Pb and Cd concentrations, and gender, age change depending on Pb concentration in erythrocytes of RTRs. Additionally, this is the first research that suggests that immunosuppressive regimen, depending on type of immunosuppressive drugs combination affects Pb concentration in erythrocytes of RTRs. Our results also confirm negative correlation between Pb and GFR which is associated with nephrotoxic properties of the mentioned metal. It seems to be crucial information for patients who use immunosuppressive drugs. However, as mentioned above, such studies should therefore be expanded.

## Figures and Tables

**Table 1 biology-10-00062-t001:** Characteristic of the study population (P1—patients under 50 years old; P2—patients over 50 years old; S—current smokers, EXS—ex-smokers, NS—never-smoking patients; IDs—immunosuppressive drugs; AM—arithmetic mean; SD- standard deviation; GFR—glomerular filtration rate; ALT—alanine transaminase; HGB—hemoglobin).

Renal Transplant Recipients (n = 115), Stable Function of Graft > 6 Months	
Parameter	Number of Patients (n)
*Age:*	
P1 < 50	43
P2 > 50	72
*Gender:*	
Man	61
Woman	54
*Smoking:*	
S	16
EXS	40
NS	43
*Number of IDs in regimen:*	
2 drugs	61
3 drugs	54
*Biochemical parameters (AM ± SD)*	
GFR	53 ± 22.41 mL/min/m^2^
ALT	22.55 ± 13.51 IU/L
HGB	8.18 ± 1.16 mmol/L
HCT	43 ± 18.22%

**Table 2 biology-10-00062-t002:** Cadmium, mercury, and lead concentrations in erythrocytes of renal transplant recipients; (P1 < 50 years old; P2 > 50 years old; AM: arithmetic mean, SD: standard deviation, Min–Max: minimum and maximum values, CV: coefficient of variation in %; * *p* < 0.05, statistically significant difference vs. MAN; ** vs. P2).

Parameters	Cd (µg/L)	Hg (µg/L)	Pb (µg/L)
Acceptable Norms in Blood:	0.3–1.2 μg/L [21]	<10 μg/L [22]	<100 μg/L [28]
*All Patients (n = 115)*			
*AM ± SD*	1.91 ± 1.48	1.85 ± 1.38	48.36 ± 29.18
*Median*	1.43	1.44	41.68
*Min–Max*	0.36–11.93	0.32–8.7	17.28–189.81
*CV*	77.31	74.95	60.35
*Woman (n = 54)*			
*AM ± SD*	2.15 ± 1.86	1.91 ± 1.49	43.51 ± 29.71
*Median*	1.43	1.61	35.18 *
*Min–Max*	0.38–11.93	0.32–8.76	17.28–189.81
*CV*	59.03	52.11	54.12
*Man (n = 61)*			
*AM ± SD*	1.71 ± 1.01	1.81 ± 1.3	52.66 ± 28.27
*Median*	1.43	1.36	44.23
*Min–Max*	0.36–4.78	0.41–8.23	19.36–163.56
*CV*	58.88	71.92	53.69
*P1 (n = 43)*			
*AM ± SD*	2.01 ± 1.47	1.55 ± 1.01	39.39 ± 28.96
*Median*	1.43	1.25	31.37 **
*Min–Max*	0.36–6.13	0.41–5.59	17.28–163.56
*CV*	73.32	64.83	73.52
*P2 (n = 72)*			
*AM ± SD*	1.86 ± 1.49	2.03 ± 1.55	53.71 ± 28.17
*Median*	1.47	1.46	48.31
*Min–Max*	0.62–11.93	0.32–8.76	18.11–189.81
*CV*	80.27	76.45	52.45

**Table 3 biology-10-00062-t003:** Cadmium, mercury and lead concentrations in erythrocytes of renal transplant recipients with division on immunosuppressive regimens (MMF: mycophenolate mofetil; CsA: cyclosporine A; Tac: tacrolimus; G: glucocorticosteroid; AM: arithmetic mean; SD: standard deviation; Min-Max: minimum and maximum values; CV: coefficient of variation in %; * *p* < 0.05, statistically significant difference vs. MMF + Tac).

Parameters	Cd (µg/L)	Hg (µg/L)	Pb (µg/L)
2 Drugs Based Regimens			
*MMF + CsA (n = 11)*			
*AM ± SD*	1.04 ± 0.80	1.95 ± 0.81	31.82 ± 13.41
*Median*	1.61	1.61	28.61 *
*Min–Max*	0.36–3.01	0.52–2.65	17.28–77.38
*CV*	54.24	43.21	52.84
*MMF + Tac (n = 50)*			
*AM ± SD*	2.02 ± 1.61	1.79 ± 1.31	50.34 ± 27.75
*Median*	1.43	1.44	46.93
*Min–Max*	0.38–11.93	0.32–8.78	17.76–163.56
*CV*	81.94	77.96	53.38
*3 Drugs Based Regimens*			
*MMF+ CsA + G (n = 6)*			
*AM ± SD*	2.02 ± 0.37	0.89 ± 0.84	31.63 ± 24.44
*Median*	1.81	1.2	24.06
*Min–Max*	1.01–3.01	0.52–1.96	17.28–77.38
*CV*	41.02	46.46	68.04
*MMF + Tac + G (n = 48)*			
*AM ± SD*	2.26 ± 1.84	1.76 ± 1.49	51.93 ± 30.45
*Median*	1.51	1.41	50.24
*Min–Max*	0.38–11.93	0.32–8.76	17.76–163.56
*CV*	87.47	86.01	55.08

**Table 4 biology-10-00062-t004:** Cadmium, mercury, and lead concentrations in blood erythrocytes of renal transplant recipients in aspect of smoking (S: smokers; EXS: ex-smokers; NS: never-smoking patients; AM: arithmetic mean, SD: standard deviation; Min-Max: minimum and maximum values; CV: coefficient of variation in %; * *p* < 0.05, statistically significant difference vs EXS, ** vs. NS).

Parameters	Cd (µg/L)	Pb (µg/L)
*S (n = 16)*		
*AM ± SD*	3.55 ± 2.55	59.24 ± 68.94
*Median*	3.25 *	47.77
*Min–Max*	0.89–11.9	19.52–163.56
*CV*	71.83	68.94
*EXS (n = 40)*		
*AM ± SD*	1.94 ± 1.1	51.98 ± 34.91
*Median*	1.58	41.61 **
*Min–Max*	0.69–6.13	18.81–189.81
*CV*	56.79	67.14
*NS (n = 45)*		
*AM ± SD*	1.29 ± 0.74	40.53 ± 17.86
*Median*	1.08	38.77
*Min–Max*	0.36–3.65	17.28–89.55
*CV*	57.26	44.08

**Table 5 biology-10-00062-t005:** Concentrations of blood Cd, Hg, and Pb (in µg/L) in other group of patients from Poland and other countries.

Country	Participants		*n*	Blood Concentration (µg/L)			References
	Age (AM)	Gender		Cd	Hg	Pb	
Acceptable Norms:				0.3–1.2 μg/L	<10 μg/L	<100 μg/L	
Our Research:				1.91	1.85	48.36	
Poland	28	F	80	0.61			Olszowski et al., 2016 [29]
Poland	20	M	30	0.56	4.28	2.48	Janicka et al., 2015 [16]
	60		30	0.56	1.78	4.48	
Korea		M	2523		5.88		Lee et al., 2017 [41]
		F	2661		4.11		
Korea	60	F + M	560		2.81		Lee et al., 2017 [9]
USA	63	F + M	3226	0.5			Kim et al., 2018 [1]
Spain	newborns			0.27	6.72	14.09	Garcia-Esquinas et al., 2013 [8]
		F		0.53	3.90	19.80	
		M		0.49	5.38	33.00	
Mexico	14–41	F	292			2.79	La-Llave Leon et al., 2015 [30]
China	5–6	F + M	855			19.30	Liu et al., 2014 [31]

## Data Availability

The data presented in this study are available on request from the corresponding author.

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
