# Peer review of "Analysis of Cadmium, Mercury, and Lead Concentrations in Erythrocytes of Renal Transplant Recipients from Northwestern Poland"

_biology, 2021, doi:10.3390/biology10010062_

Round 1

Reviewer 1 Report

General evaluation

This paper addressed a very important topic for renal transplant recipients and the care team. The Immunosuppressive Regimen choice has more consequences that should be considered in future. This is a relevant research topic .

Following elements can be considered to strengthened the paper further:

Title

The title is very vague. Please add a hint to show what kind of study can be expected.

Abstract

The abstract is well structured. There are a lot of abbreviation that have to be introduced. Somewhere in the article the abbreviation are explained, however often not when it is mentioned the first time.

Keywords

ok

Introduction

The introduction and the discussion are mixed. In the introduction I expected to read more about what is known and what are the regional differences in healthy people.

Please add a theoretical model /framework to your hypotheses.

Results

Results are clear. A statistician might have to look if these analyses are right – it is not in my expertise. However the comparison is between a very small sample size and a small sample. Did the authors really use the right test for this sample size?

Please add a description of the renal transplant patient. How long are they already transplanted?

Material and Methods

Material and Methods are clear. What did you do with missing data?

Data plausibility, how did you check that data imput was correct? Please add. Please add that you did descriptive analyses.

Discussion

Discussion is ok, I would write some of these information in the introduction.

What were the limitations of this study? What do you recommend for the clinical practice? What do you suggest as future study to be done in renal transplant recipients?

What will be the consequences for health policies?

Figures and Tables

Figure 1: Please use a normal moderate figure. Too fancy. Better a descriptive table of the population, adding some more transplant information. Do figure 2 and 3 add relevant Information? What about Table 4? I would delete this 3.

Table 5 – Where the studies mentioned in table 5 all RCT?, valid? Reliable?... Please add some more information.

Author Response

Dear Reviewer #1:

Thank you for your advice and constructive comments concerning our manuscript entitled, “ Analysis of cadmium, mercury and lead concentrations in erythrocytes of renal transplant recipients from northwestern Poland”. We have carefully considered your suggestions, and have revised our manuscript accordingly; we hope that these changes meet with your approval. We have highlighted our changes, as red colour.

Reviewer #1:

This paper addressed a very important topic for renal transplant recipients and the care team. The Immunosuppressive Regimen choice has more consequences that should be considered in future. This is a relevant research topic .

Following elements can be considered to strengthened the paper further:

Title

The title is very vague. Please add a hint to show what kind of study can be expected.

Thank you for that comment. We have changed the title, starting with „Analysis” that reflect the type of the manuscript.

Abstract

The abstract is well structured. There are a lot of abbreviation that have to be introduced. Somewhere in the article the abbreviation are explained, however often not when it is mentioned the first time.

Thank you for that comment. We have explained the abbreviation.

Keywords

ok

Introduction

The introduction and the discussion are mixed. In the introduction I expected to read more about what is known and what are the regional differences in healthy people.

Please add a theoretical model /framework to your hypotheses.

Thank you for that comment. We have improved the introduction. We have added some information regarding the values of Cd, Pb and Hg from other regions. We have also improved the hypothesis.

Results

Results are clear. A statistician might have to look if these analyses are right – it is not in my expertise. However the comparison is between a very small sample size and a small sample. Did the authors really use the right test for this sample size?

Thank you for that comment. We used nonparametric test – Umann-Whitney to check if there are any differences between the tested groups of patients. The numer of patients for the test is correct. Additionally, we have checked some more correlation – Spearman correlation, to check if there are any relationships between metal concentration and age, gender, GFR, ALT,HGB.

Please add a description of the renal transplant patient. How long are they already transplanted?

Thank you for that comment. We have added this information.

Material and Methods

Material and Methods are clear. What did you do with missing data?

Dear Reviewer, we had all results and we had all 3 metals concentrations.

Data plausibility, how did you check that data imput was correct? Please add. Please add that you did descriptive analyses.

Thak you for that comment. We used reference material to check the method. We have highlighted it in red.

Germanium (PerkinElmer Pure, PerkinElmer Life and Analytical Sciences, USA) was used as an internal standard and ClinChek Plasma Control Level I (Recipe, Germany) was used as a reference material, measured after each of the six samples: where the difference of the reference material measurements exceeded 5%, the entire series was repeated.

Discussion

Discussion is ok, I would write some of these information in the introduction.

Thank you for that constructive comment. We have changed the introduction and we have added some more information.

What were the limitations of this study? What do you recommend for the clinical practice? What do you suggest as future study to be done in renal transplant recipients?

What will be the consequences for health policies?

We are grateful for that comment. We have added the limitations and recomendation for the future.

Figures and Tables

Figure 1: Please use a normal moderate figure. Too fancy. Better a descriptive table of the population, adding some more transplant information. Do figure 2 and 3 add relevant Information? What about Table 4? I would delete this 3.

Thank you for that comment. We have changed the Figure 1- on Table 1. We have added some more data in it, regarding transplantation – biochemical parameters. We also deleted Figures 2 and 3 and Table 4. Instead Table 4, we described our results, since as mentioned above, we additionally counted Spearman correlation between metals and age, gender, GFR, HGB, ALT.

Table 5 – Where the studies mentioned in table 5 all RCT?, valid? Reliable?... Please add some more information.

Thank you for that comment. We have added some information about the data and patients in Table 5, however, we are very sorry, but to the best of our knowledge, there are no results regarding toxic metals in renal transplant recipients. Therefore, we added acceptable norms in blood and our results for the better view and comparison of the concentrations od metals in different geaografical regions.

We wish to thank the Reviewer for extremely helpful comments. We hope that this manuscript is acceptable for publication in Biology. Further researches are needed in the following subject and need to be expanded and we plan to do it.

Reviewer 2 Report

Overall nice manuscript and great addition to the literature. 

To make it great contribution to the literature, I will recommend: 

  1. Adding kidney function data based on the levels of Cd, Pb, Hg. 
  2. Dose the increasing level of Cd, Pb or Hg was associated with increased risk of graft loss or patient death?
  3. Was there any difference in rejection? 

Author Response

Dear Reviewer #2:

Thank you for your advice and constructive comments concerning our manuscript entitled, “ Analysis of cadmium, mercury and lead concentrations in erythrocytes of renal transplant recipients from northwestern Poland”. We have carefully considered your suggestions, and have revised our manuscript accordingly; we hope that these changes meet with your approval. We have highlighted our changes, as green colour.

Reviewer #2:

Overall nice manuscript and great addition to the literature.

To make it great contribution to the literature, I will recommend:

Adding kidney function data based on the levels of Cd, Pb, Hg.

Thank you for that constructive comment. We have added biochemical parametres in table 1, regarding the function of transplanted organ. We have also checked if there are any relationships bewteen Cd, Pb and Hg and age, gender, GFR, HGB, HCT, ALT.

Dose the increasing level of Cd, Pb or Hg was associated with increased risk of graft loss or patient death?

Thank you for that comment. We have checked the corelation as mentioned above. There was negative significant correlation between Pb and GFR, what we have added in results and discussion. It was the confirmastion of toxic properties of Pb.

Was there any difference in rejection?

Thank you Sir/Madam for that comment. We did not have any patients in the current study with rejection. The mean biochemical parameters were correct:

GFR

53±22.41

ALT

22.55±13.51

HGB

8.18±1.16

HCT

43±18.22

We wish to thank the Reviewer for extremely helpful comments. We hope that this manuscript is acceptable for publication in Biology. Further researches are needed in the following subject and need to be expanded and we plan to do it.

Reviewer 3 Report

Paper reports erythrocyte concentrations of some toxic elements (Cadmium, Mercury and Lead) in a group of renal transplanted patients. Obtained values have been compared between males and females, age > or < 50yrs, smokers or non smokers and antirejection therapy regimens. Authors in previous papers from same patients reported data on serum or erythrocytes content of various bioelements (Se, Zn, Cu, Fe, As, Se).

Some notes:

1) In the Introduction section of actual paper they discuss human toxicity of Cd, Hg and Pb but the reason to choose red blood cells for analysis is not clearly presented. Do these heavy metals accumulate inside red blood cells ? is heavy metal erythocytes value a marker of intracellular/tissue deposition ? Some physiopathological element of cellular toxicity from these heavy metals should be added to support reader's understanding of the problem. In addition it should be usefull to relate erythrocytes values to blood levels for the same element. Moreover values (either blood or RBCs)  from a normal population from region/country should be used as reference.  

2) in pag 2/13 Results are presented as Section n° 2 while Matherial and Methods are at pag 9/13 as Section 4 and Conclusions are at page 10/13 as Section 5. It results that Section 3 is missing and suggestion is to present a corrent sequence and to set, as usual, Matherial and Methods before Results.

3) In the description of study population (pag 9/13, line 261) Authors refer to previous papers but, for clarity, descriptive data from group on study should be reported.  Data within Figure 1 should be considered not a result but a part of the population description inside Material and Method Section (figure 1 could therefore be eliminated).    

4) At pag 3/13 section 2.1 lines 92,93,94 report details of performed statistical analysis and therefore would be better positioned inside section 4.3 statistical analysis at pag 10/13.

5) In pag4/13 Table 1 could include values from a normal ealthy control group as suggested in note 1.

6) pag 4/13, section 2.2, lines 120,121: number of pts on each immunosuppressive regimen could be reported within descriptive data of pts on study within Material and Method section; correlations between each single heavy metal RBC value and drug concentration should be searched and discussed.

7) figure 2 and figure 3, pag 6-7/13, report on ordinate axes "Blood..concentration" while figures caption correctly reports "blood erythrocytes". Text on both ordinates needs to be changed.

8) Pag 7/13 section 2.4, line 177: text could be expanded reporting data from Table 4 and therefore the same Table 4 could be removed.   

9) pag 8/13 line 188 and Table 5: Reporting blood concentrations derived from other studies the AAs illustrate a wide range of results for each heavy metal (Cd 0.27-0.61; Hg 1.78-6.72; Pb 2.48-33.00) and this emphasizes the need to have reference data on the normal population of the same area, obtained with the same method  (already noted in note 1) 

10) From pag 8/13, line 209 to pag 9/13 line 259: Text presented here appears as a discussion section and it should be made out as such.   

Author Response

Dear Reviewer #3:

Thank you for your advice and constructive comments concerning our manuscript entitled, “ Analysis of cadmium, mercury and lead concentrations in erythrocytes of renal transplant recipients from northwestern Poland”. We have carefully considered your suggestions, and have revised our manuscript accordingly; we hope that these changes meet with your approval. We have highlighted our changes, as blue colour.

Reviewer #3:

aper reports erythrocyte concentrations of some toxic elements (Cadmium, Mercury and Lead) in a group of renal transplanted patients. Obtained values have been compared between males and females, age > or < 50yrs, smokers or non smokers and antirejection therapy regimens. Authors in previous papers from same patients reported data on serum or erythrocytes content of various bioelements (Se, Zn, Cu, Fe, As, Se).

Some notes:

  • In the Introduction section of actual paper they discuss human toxicity of Cd, Hg and Pb but the reason to choose red blood cells for analysis is not clearly presented. Do these heavy metals accumulate inside red blood cells ? is heavy metal erythocytes value a marker of intracellular/tissue deposition ? Some physiopathological element of cellular toxicity from these heavy metals should be added to support reader's understanding of the problem. In addition it should be usefull to relate erythrocytes values to blood levels for the same element. Moreover values (either blood or RBCs) from a normal population from region/country should be used as reference. 

Thank you very much for that comment. We have added the information in the introduction and describe whythe analysis of toxic metals levels is reasonable in erythrocytes. We have also added the information of cellular toxicity of toxic metals. The ICP-MS method allows for an extremely accurate and precisely examination of the concentration of metals in the blood,  because heavy metals have an affinity for the membrane of erythrocytes, their concentrations in the whole blood are not generally tested, as they would be a reflection of the concentration in erythrocytes anyway. We are planning another project where we will compare 2 material types – i.e. Selenium level in serum and in blood.

  • in pag 2/13 Results are presented as Section n° 2 while Matherial and Methods are at pag 9/13 as Section 4 and Conclusions are at page 10/13 as Section 5. It results that Section 3 is missing and suggestion is to present a corrent sequence and to set, as usual, Matherial and Methods before Results.

Thank you very much for that comment. We have changed the Material and Methods section before Results.

  • In the description of study population (pag 9/13, line 261) Authors refer to previous papers but, for clarity, descriptive data from group on study should be reported. Data within Figure 1 should be considered not a result but a part of the population description inside Material and Method Section (figure 1 could therefore be eliminated).   

Thank you for that comment. We have changed the Figure 1 –instead we have added table 1 with descriptive information- buochemical parameters reflecting the function of the graft. We have put this Table 1 in Material and Methods section.

  • At pag 3/13 section 2.1 lines 92,93,94 report details of performed statistical analysis and therefore would be better positioned inside section 4.3 statistical analysis at pag 10/13.

Thank you very much for that comment. We have put these sentences in Statistical analysis.

  • In pag4/13 Table 1 could include values from a normal ealthy control group as suggested in note 1.

Thank you for that comment. We have added „acceptable norms” Table 2.

  • pag 4/13, section 2.2, lines 120,121: number of pts on each immunosuppressive regimen could be reported within descriptive data of pts on study within Material and Method section; correlations between each single heavy metal RBC value and drug concentration should be searched and discussed.

Thank you for that comment. We have added the numer of patients who are on 2 drugs regimen and 3 drugs regimmen. The numer of patients on detailed immunosuppressive regimen is included in Table 3. We have also checked the correlation between metals levels in erythrocytes and drugs concentration using Spearman correlation, and no significant correlation was noted. We have added this information in 3.4. Section.

  • figure 2 and figure 3, pag 6-7/13, report on ordinate axes "Blood..concentration" while figures caption correctly reports "blood erythrocytes". Text on both ordinates needs to be changed.

Thank you for that comment. We have deleted these figures, since they repeat the values present in the Table 4.

  • Pag 7/13 section 2.4, line 177: text could be expanded reporting data from Table 4 and therefore the same Table 4 could be removed.

Thank you for that comment. We have removed Table 4.

  • pag 8/13 line 188 and Table 5: Reporting blood concentrations derived from other studies the AAs illustrate a wide range of results for each heavy metal (Cd 0.27-0.61; Hg 1.78-6.72; Pb 2.48-33.00) and this emphasizes the need to have reference data on the normal population of the same area, obtained with the same method (already noted in note 1)

Thank you for that constructive comment. We have added „acceptable norms” in Table 5. We ahve also added our results regarding metals concentrations in erythrocytes to make it clear and to enable to compare our results, from Poland with other region of Poland and with other results in different countries. It is the limitation of our research, that to our best knowledge, we have not found any results regarding blood concentrations of Cd, Pb and Hg in RTRs. We have also added the limitation at the end of the manuscript.

  • From pag 8/13, line 209 to pag 9/13 line 259: Text presented here appears as a discussion section and it should be made out as such.

Thank you for that comment. We have separated Discussion section clearly.

We wish to thank the Reviewer for extremely helpful comments. We hope that this manuscript is acceptable for publication in Biology. Further researches are needed in the following subject and need to be expanded and we plan to do it.

Reviewer 4 Report

Aleksandra Wilk et al. in this paper evaluate the intra-erythrocyte concentration of lead, mercury and cadmium in a population of kidney transplant recipients.

The authors describe an association between a high intra-erythrocyte concentration of mercury and female sex and high concentration of lead and male sex.

Furthermore, they find a higher lead intra-erythrocyte concentration in patients older than 50 years and in patients treated with Tacrolimus and MMF.

 Lastly, they describe a significant correlation between intra-erythrocyte concentration of cadmium and lead.

The study design shows numerous points that should be reviewed.

My suggestion is to expand the experimental design including:

  1. a control group of patients in hemodialysis (pre-transplant control) belonging to the same geographical area and with similar demographic characteristics
  2. a group of healthy subjects belonging to the same geographical area and comparable in all except for immunosuppression
  3. a group of heart or liver transplant recipients without renal failure

Furthermore, the authors, in order to demonstrate that immunosuppressant drugs can lead to elevated intra-erythrocyte concentrations of heavy metals, should exclude other potential causal or associated factors by planning a multivariate analysis that consider confounder variables as gender, age, GFR, hematocrit, smoking status, use of erythropoietin, transplant age.

Minor points:

The tables and figures should be improved and the discussion reviewed.

Lastly, the authors should justify why they chose a 50-year-old cut-off to perform some statistical analysis.

Author Response

Dear Reviewer #4:

Thank you for your advice and constructive comments concerning our manuscript entitled, “ Analysis of cadmium, mercury and lead concentrations in erythrocytes of renal transplant recipients from northwestern Poland”. We have carefully considered your suggestions, and have revised our manuscript accordingly; we hope that these changes meet with your approval. We have highlighted our changes, as orange colour.

Reviewer #4:

Aleksandra Wilk et al. in this paper evaluate the intra-erythrocyte concentration of lead, mercury and cadmium in a population of kidney transplant recipients.

The authors describe an association between a high intra-erythrocyte concentration of mercury and female sex and high concentration of lead and male sex.

Furthermore, they find a higher lead intra-erythrocyte concentration in patients older than 50 years and in patients treated with Tacrolimus and MMF.

 Lastly, they describe a significant correlation between intra-erythrocyte concentration of cadmium and lead.

The study design shows numerous points that should be reviewed.

My suggestion is to expand the experimental design including:

1/ a control group of patients in hemodialysis (pre-transplant control) belonging to the same geographical area and with similar demographic characteristics

a group of healthy subjects belonging to the same geographical area and comparable in all except for immunosuppression

a group of heart or liver transplant recipients without renal failure

Thank you for that constructive comment. We are sorry, however, we do not have such a material that you have mentioned above. It would be great to expand our research and we have that plan, however, firstly, we wanted t Focus on renal transplant recipients, since this is the final step of the current project. Previously we have described the selenium, iron, zinc and copper concentrations in serum of RTRs: Wilk A, Szypulska-Koziarska D, Marchelek-Mysliwiec M, Glazek W, Wiszniewska B: Serum Selenium, Iron, Zinc, and Copper Concentrations in Renal Transplant Recipients Treated with Mycophenolate Mofetil. Biol Trace Elem Res 2020.

We also described arsenic and selenium concentration in erythrocytes of RTRs: Arsenic and Selenium Profile in Erythrocytes of Renal Transplant Recipients. Biological trace element research 2020, 197(2).

We would like to end this with closing the project with the toxic metals. We are grateful for you comment and we plan to expand our reseacrh on heart, liver graft and hemodialyzed patients. Unfortunately, the situation in our hospital is not proper for collecting material (Covid-19 situation).

2/ Furthermore, the authors, in order to demonstrate that immunosuppressant drugs can lead to elevated intra-erythrocyte concentrations of heavy metals, should exclude other potential causal or associated factors by planning a multivariate analysis that consider confounder variables as gender, age, GFR, hematocrit, smoking status, use of erythropoietin, transplant age.

Thank you for your comment. We improved our statistical analysis and we have checked the correlation between the parameters, according to Your suggestions. We have added this information in methods, results section, and discussion.

Minor points:

3/ The tables and figures should be improved and the discussion reviewed.

Thank you for that comment.We have changed the Tables: Instead Figure 1 we have created Table 1 with patients status information- regarding the function of the graft. We have deleted Figures 2 and 3, since they repeated data from Table 4.

4/ Lastly, the authors should justify why they chose a 50-year-old cut-off to perform some statistical analysis.

Thank you for that comment. We have justified why we chose a 50-year-old cut-off to perform some statistical analysis in discussion section.

We wish to thank the Reviewer for extremely helpful comments. We hope that this manuscript is acceptable for publication in Biology. Further researches are needed in the following subject and need to be expanded and we plan to do it.

Round 2

Reviewer 3 Report

The paper has been adapted to reviewers suggestions and current version appears much more worthy of publication. Still text would deserve a revision of English style.

I add some notes to further improve paper:

1) Material and Methods section, pag3/15:  In line 135 is presented a cut-off of 50 years used in patients data analysis and the same is presented in Table 1; the reason for this selection is only presented in Discussion section, pag 9/15, lines 313-316. I suggest to move this part of the discussion and related references from Discussion to Introduction section as it would improve the reader's understanding of the choices made in grouping patients.

2) Table 1: Biochemical parameters need an indication of the unit of measurement or within table or in caption.

3) Table 2, page 6/15: Within table acceptable norms are reported but there is a need to indicate in caption or within same table if they are referred to RBCs  (as data in blood are quite different and already reported in the Introduction). Moreover for Hg and Pb the reported values for acceptable norms need to be more detailed with the indication if > or < than the indicated value. Finally these reported acceptable norms should be referenced.

4) Table 3, page 7/15: Within table in 5th line Median value for Pb in MMF+CsA group reaches significance. To improve data readability I suggest to leave only the mark * and to transfer the rest of the text into the caption.

5) Table 4, page 8/15: I suggest to treat median values of Cd in smokers as indicated in previous note.

6) References section, pages 13-14/15: references 15, 17, 19, 20,21,22, 26 simply report AAs initials: this is unaccepatble. Moreover reference 18 lacks Author and Journal names. I suggest to carefully review bibliography.        

Author Response

Dear Reviewer #3:

Thank you for your advice and constructive comments concerning our manuscript entitled, “ Analysis of cadmium, mercury and lead concentrations in erythrocytes of renal transplant recipients from northwestern Poland”. We have carefully considered your suggestions, and have revised our manuscript accordingly; we hope that these changes meet with your approval. We have highlighted our changes, as red colour.

Reviewer #3:

The paper has been adapted to reviewers suggestions and current version appears much more worthy of publication. Still text would deserve a revision of English style.

I add some notes to further improve paper:

1) Material and Methods section, pag3/15:  In line 135 is presented a cut-off of 50 years used in patients data analysis and the same is presented in Table 1; the reason for this selection is only presented in Discussion section, pag 9/15, lines 313-316. I suggest to move this part of the discussion and related references from Discussion to Introduction section as it would improve the reader's understanding of the choices made in grouping patients.

Thank you for that comment. We have changed this and we have moved this lines to the Introduction.

2) Table 1: Biochemical parameters need an indication of the unit of measurement or within table or in caption.

Thank you very much. We have added the units in Table 1.

3) Table 2, page 6/15: Within table acceptable norms are reported but there is a need to indicate in caption or within same table if they are referred to RBCs  (as data in blood are quite different and already reported in the Introduction). Moreover for Hg and Pb the reported values for acceptable norms need to be more detailed with the indication if > or < than the indicated value. Finally these reported acceptable norms should be referenced.

Thank you. We have indicated < sign in norms.

4) Table 3, page 7/15: Within table in 5th line Median value for Pb in MMF+CsA group reaches significance. To improve data readability I suggest to leave only the mark * and to transfer the rest of the text into the caption.

Thank you for that comment. We have changed this.

5) Table 4, page 8/15: I suggest to treat median values of Cd in smokers as indicated in previous note.

Thank you very much. We have improved this as in Table 3.

6) References section, pages 13-14/15: references 15, 17, 19, 20,21,22, 26 simply report AAs initials: this is unaccepatble. Moreover reference 18 lacks Author and Journal names. I suggest to carefully review bibliography.    

Thank you. We have checked References and we have corrected this section.

We wish to thank the Reviewer for extremely helpful comments. This help made the article much more clear and understandable. We hope that this manuscript is acceptable for publication in Biology.

Reviewer 4 Report

The authors have made only partial changes to this study
being a project already concluded, however the revised version
has increased the value of the study.
I have no further suggestions for the authors.

Author Response

Dear Reviewer #4:

Thank you for your positive opinion concerning our manuscript entitled, “ Analysis of cadmium, mercury and lead concentrations in erythrocytes of renal transplant recipients from northwestern Poland”.  Your help made the article much more clear and understandable.
